# Functional trait identity regulates productivity better than tree diversity and structural complexity in subtropical mixed-species forests

**Xiaoyu Chen°, Meng Xiang°, Lan Yao‡, Xuru Ai‡, Jiang Zhu‡, Qiuju Guo‡, Keyan Zhang‡, Qiang Zhang** ©*

Hubei Key Laboratory of Biologic Resources Protection and Utilization, Hubei Minzu University, Enshi, China,

° Xiaoyu Chen and Meng Xiang contributed equally to this work.
‡ LY, XA, JZ, QG and KZ also contributed equally to this work.
* zq34011456@126.com

## Abstract

Subtropical forests play an important role in global carbon cycle and in mitigating climate change. Understanding the relationship between multiple diversity and ecosystem function is crucial for protecting and managing forests. Here we used forest inventory data from a 6-hectare sample plot in natural evergreen deciduous broadleaf mixed forest systems for the years 2016 and 2021. We analyzed the effects of multiple aspects of diversity and topographic factors on forest productivity using multiple causal analyses. We found that *Fagaceae* was the primary contributor to productivity in the forest stand. Elevation, slope, and convexity showed no significant effects on productivity. Structural complexity (stand density, large-diameter trees and tree-size variation) was significantly positively correlated with productivity. Taxonomic and functional diversity indices were weakly correlated with productivity. Specifically, forest productivity was enhanced by traits associated with greater maximum height and lower wood density. Community-weighted mean traits were the most strongest predictor of productivity relative to other variables. Within this forest stand, the mass-ration hypothesis appeared to be more influential on forest productivity compared to the complementarity and selection effects. By integrating multiple drivers of forest ecosystem functioning, our study provides critical system-level insights needed to predict the potential consequences of regional changes in forest diversity, composition, structure and function.

## 1. Introduction

Biodiversity and ecosystem functioning relationships have been extensively researched over the past three decades [1–4]. These studies consistently demonstrate that taxonomic diversity promotes forest productivity and biomass accumulation at both local and global scales [5–7]. Among them, two non-exclusive

**Data availability statement:** Species information is available at https://www.iplant.cn and http://www.try-db.org.

**Funding:** This work was supported by the National Key Research and Development Program of China (2023YFE0112800), PhD Start-Up Fund Project (RZ2400000541) and Open Fund Project of Hubei Key Laboratory of Biologic Resources Protection and Utilization (KYPT012312). The plant collection and use was in accordance with all the relevant guidelines. The funders had no role in study design, data collection and analysis, decision to publish, or preparation of the manuscript.

**Competing interests:** The authors have declared that no competing interests exist.

mechanisms explain the positive relationships between biodiversity and ecosystem functioning. The complementarity effect suggests that higher tree diversity in plant communities enhances resource-use efficiency, thereby boosting forest productivity [8], Early on, complementarity effects were often quantified by taxonomic diversity indices, such as species richness [8,9]. Additionally, the selection effect states that more diverse communities are likely to contain more species with high performance in ecosystem functions [8]. To some extent, the selection effects also supports the 'mass-ration hypothesis' [10], which postulates that the traits of dominant species largely determine ecosystem function [10], often quantified by community-weighted mean of functional trait values [10–12]. Despite extensive research on these mechanisms, their relative importance varies substantially among different forest types [13–15].

The intricate structure of forest stands plays a crucial role in determining ecosystem functioning [16,17]. Structural complexity not only encapsulates the spatial distribution of communities but also serves as an indicator of overall biodiversity within these communities, the inherent variances among different species, coupled with asymmetric competition between homogeneous and heterogeneous individuals, contribute to the diverse spatial allocation of trees, thereby enhancing resource utilization efficiency and productivity [17,18]. In practice, higher densities promote canopy packing, improving light capture and utilization by co-existing species, thereby increasing biomass and production [18,19]. The observed positive correlation between tree-size inequality and biomass (or productivity) may be attributed to the niche complementarity effect, which posits that forest communities exhibiting greater structural complexity can enhance resource use efficiency, thereby increasing forest functioning [7,18,20,21]. Compared to small and medium-sized trees, larger-diameter trees play a crucial role in forest structure, functionality, and diversity by competing for sunlight [22–26]. Considering the crucial role that large trees play in the global carbon cycle [27], they have become a focal point in the study of forest ecosystem under global climate change [28–30]. Simultaneously, research has demonstrated that both taxonomic and functional diversity exert direct or indirect influences on forest biomass and productivity by way of the intricate complexity of stand structure [20,31,32].

Besides taxonomic diversity, functional trait composition and structural complexity, several other abiotic factors (e.g., elevation, slope, soil nutrients, etc) have also been recognized as direct or indirect influences forest biomass and productivity [20,31,33,34]. For instance, topography (i.e., elevation) is a key spatial driver of biodiversity, life attributes, functional composition and stand structure, as it constrains micro- and macro-climatic conditions, soil nutrients and hydraulic conditions for tree growth [14,35–37]. Previous studies have shown that the negative effect of elevation on biomass is attributed to lower temperatures, reduced soil water content, and shallower soils with increasing altitude [38–40]. There are also related studies that demonstrate how tree diversity, structure and functional trait identity promote stand biomass along elevational gradients in subtropical forests of Southern China [41]. Therefore, many abiotic factors must be considered when studying the relationship between biodiversity and ecosystem function.

Species-rich and structurally complex forest ecosystems play vital roles in maintaining biodiversity and regulating the carbon cycle [42–44]. Mixed-species forests with greater tree species richness and structural complexity enhance resource use efficiency, thereby increasing forest productivity and biomass accumulation [45–47]. Understanding how biotic and abiotic factors influence forest productivity and elucidating causal mechanisms can strengthen and improve forest management and conservation strategies [48,49]. In this study, we investigated the main drivers regulating wood productivity and their underlying mechanisms in natural evergreen deciduous broad-leaf mixed forest systems by analyzing various biotic and abiotic factors. Here, we used forest inventory data from a 6-ha forest plot collected between 2016 and 2021. Specifically, we address the following scientific questions. (1) In mixed-species communities, which species constitute the most to overall productivity, and how do their functional traits of these species influence productivity? (2) How do tree diversity, functional trait identity, structural complexity and topography influence forest productivity, and what are the causal relationships and potential underlying mechanisms?

## 2. Materials and methods

### 2.1 Study site

Our study was conducted at the Jinzishan Forest Field (109°2'58'E, 30°17'17'N) in Lichuan City, Hubei Province. The region has a typical monsoon humid climate, and is located at an altitude of 1,000–1,760 m. The average annual air temperature is 12.7°C, and the mean annual precipitation is 1, 450 mm, respectively. Soil is mainly upland yellow and brown soil, with soil pH of 5.0–6.0. The main dominant vegetation type is this evergreen deciduous broad-leaved mixed forest. Tree canopy is mostly tall deciduous trees, the middle and lower layers are mainly evergreen trees. The maximum tree height and diameter at breast height (DBH, measured at 1.3 m height) reached 25.6 m and 64.8 cm, respectively. The main deciduous species are *Castanea henryi*, *Carpinus fargesiana*, *Quercus serrata* and *Bothrocaryum controversum*, and the main evergreen species are *Nothopanax davidii*, *Camellia cuspidata* and *Eurya alata*.

### 2.2. Forest inventory and collection of plant functional traits

We established a 6-hectare (200 × 300 m) permanent plot using a whole-station instrument, divided into 150 20 × 20 m subplots. A first tree census was performed in October 2016 while a second one was performed in October 2021. All freestanding woody stems ≥5 cm in DBH were identified, tagged, measured, and mapped using standardized census protocols. For all tree species, we collected two key functional trait indicators to reflect plant light acquisition and growth strategies: plant maximum height (MH, available at https://www.iplant.cn) and wood density (WD, available at http://www.try-db.org). Missing trait values were replaced with data from closely related species of the same genus.

### 2.3. Variables used in analyses

We estimated standing biomass (Mg) for each species using DBH-based allometric biomass equations of this region for the branches ($W_B = 0.027DBH^{2.428}$), stems ($W_S = 0.122DBH^{2.263}$), leaves ($W_L = 0.024DBH^{2.029}$), and roots ($W_B = 0.041DBH^{2.361}$), respectively. That the allometric relationships were taken from "China's Forest Ecosystem Carbon Storage-Biomass Equation" [50]. Forest productivity was estimated as the annual biomass increment of all trees reaching at least 5 cm DBH from the first census (T0) to the last census (T1), and divided by the time interval. In each subplot, we calculated the sum of the productivity of all trees.

We used three taxonomic diversity indices, including species richness, Shannon-Wiener index, and Pielou index [51]. We used the Rao's Q diversity index and two community-weighted mean (CWM) trait values ($CWM_{MH}$ and $CWM_{WD}$) to indicate functional trait diversity and composition. Functional trait diversity and CWM indices were calculated using R package 'FD' [52]. In addition, we used stand density, maximum DBH (DBHmax) and standard deviation of DBH (Stdv DBH) to measure stand structural complexity [14,41].

We used elevation, slope, and convexity to represent terrain undulation, steepness, and curvature, providing a comprehensive assessment of topographic features. We calculate the slope and convexity based on the elevation of each subplot. The slope is calculated by the difference between the highest and the lowest points of each subplot, and then obtained by using the trigonometric function. The convexity is calculated by subtracting the average elevation of the 8 adjacent subplots from the elevation of the central subplot.

### 2.4 Statistical analyses

Within the 6-ha plot, we analyzed the relative contribution of different species to overall productivity. We used a general linear modeling (GLM) and pairwise pearsons correlation coefficients to analyze the relationship between each species' functional traits and abundance (defined as the number of individuals of each species) with their productivity. For $20 \times 20$ m subplot level, we analyzed the pairwise relationship between predictor variables and productivity. Before fitting subsequent models, all metrics were standardized by calculating z-scores to aid in model convergence. To remove the multicollinear variables, we selected non-collinear with variables with a variance inflation factor (VIF) of less than 3. These variables were then excluded from subsequent analyses. We calculated the VIF using the R package 'CAR' [53]. We used multiple regression modeling (MRM) to examine the effects of multiple predictors on biomass and productivity. We selected the best model results using the Akaike information criterion (AIC) by considering the lowest AIC and number of predictors[54], as implemented in the R package 'MuMIn' [55].

We utilized structural equation modeling (SEM) to evaluate the direct and indirect drivers of wood productivity according to a preconceived hypothesis regarding causal relationships. To create a composite variable for each hypothetical determinant, we applied the principal component analysis (PCA) with the highest overall correlation with each type of variable in modelling and selected the PC axes [34]. This decision did not influence model fitting or the magnitude of path coefficients, but solely affected the sign. The causal direction of influence of individual predictors on productivity could still be determined by comparing the sign of its bivariate correlation. We explored a full model that included all possible pathways and sequentially removed pathways that had minimal impact (<0.01) and did not attain a significant level of $P < 0.05$ to obtain the final model. Maximum likelihood estimation was used to estimate path coefficients, and the goodness of fit statistics of the final model was evaluated against the the following four criteria: (i) the ratio of Chi-square test ($\chi^2$; the model has a good fit when $0 \leq \chi^2/df \leq 2$ and $0.05 < P \leq 1.00$), (ii) root mean square error of approximation (RMSEA; the model has a good fit when $0 \leq RMSEA \leq 0.05$ and $0.10 < P \leq 1.00$), (iii) goodness-of-fit index (GFI, the model has a good fit when $0.95 \leq GFI \leq 1$), (iv) comparative goodness-of-fit index (CFI, the model has a good fit when $0.97 \leq GFI \leq 1$). The SEM were fitted using lavaan package [56]. All statistical analyses were performed in the R statistical software package, version 4.2.1 [57].

## 3. Results

Out of the 120 species, stand productivity of *C. henryi* is the largest within a 6-ha plot (11.28 Mg, 35.83%), followed by *P. lasiocarpa* (2.12 Mg yr$^{-1}$, 6.73%), *B. controversum* (2.11 Mg yr$^{-1}$, 6.71%) and *C. fortunei* (1.98 Mg yr$^{-1}$, 6.31%), these four species accounted for 55.58% (Fig 1a). Stand productivity of *Fagaceae* is the largest (12.60 Mg yr$^{-1}$, 40.04%) in this forest plot, followed by *Cupressaceae* (3.65 Mg yr$^{-1}$, 11.60%) and *Cornaceae* (3.32 Mg yr$^{-1}$, 10.56%) (Fig 1b). Species abundance was significantly and positively correlated with productivity ($R^2 = 0.23$, $P < 0.001$; Fig 1c). The analysis of the relationship between functional trait values and productivity of each species in this plot showed that MH was significantly and positively correlated with productivity ($R^2 = 0.22$, $P < 0.001$; Fig 1d), whereas WD was negatively correlated with productivity ($R^2 = 0.05$, $P < 0.05$; Fig 1e).

At the $20 \times 20$ m subplot level, productivity ranged from 0.53 to 2.82 Mg yr$^{-1}$. The GLM revealed that three topographic variables exhibited weak correlations with productivity ($R^2 < 0.01$, $P > 0.05$; Fig 2a-c). For taxonomic diversity indices, species richness was not significantly correlated with productivity ($R^2 = 0.01$, $P = 0.16$; Fig 2d). The Shannon-Wiener index was

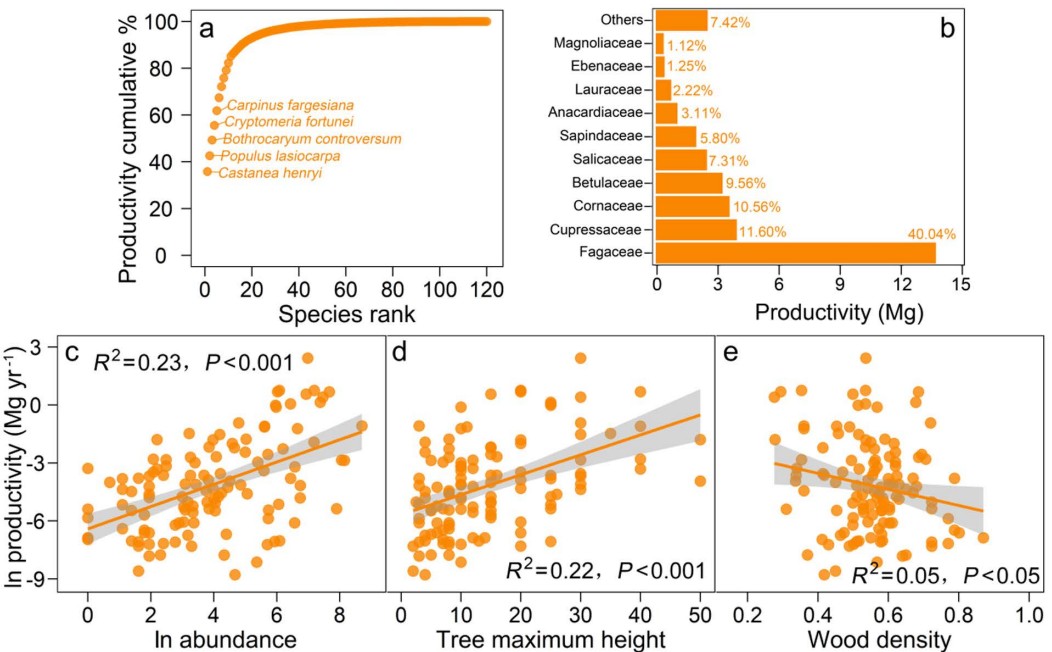

**Fig 1. Species-level (family-level) productivity ranks and their relationships with species productivity, abundance and two functional traits (tree maximum height and wood density).**

significantly negatively correlated with productivity ($R^2 = 0.05$, $P < 0.01$; Fig 2e). The Pielou index was no significantly correlated with productivity ($R^2 = 0.00$, $P = 0.88$; Fig 2f). Remarkably, functional trait diversity (Rao's Q index) was not significant related to productivity ($R^2 = 0.00$, $P = 0.86$; Fig 2g). $CWM_{MH}$ was positively related to productivity ($R^2 = 0.10$, $P < 0.001$; Fig 2h). $CWM_{WD}$ was negatively correlated with productivity ($R^2 = 0.13$, $P < 0.001$; Fig 2i). Among structural variables, stand density, DBHmax, Stdv DBH were significantly positively related to productivity ($R^2 > 0.08$, $P < 0.001$; Fig 2j-l).

The standardized final MRM results showed that stand density and Stdv DBH significantly and positively affected stand productivity ($P < 0.001$). $CWM_{MH}$ significantly and positively affected stand productivity ($P < 0.001$), while $CWM_{WD}$ had a significant negative effect on productivity ($P < 0.001$). Rao's Q index showed a weak, non-significant negative effect on productivity ($P = 0.05$) (Table 1).

The SEM results showed that multiple predictor factors directly and indirectly explained 26% of productivity. Among these, CWM traits had the strongest negative direct effect on productivity (direct effect = −0.54), followed by a direct positive effect on structural diversity (direct effect = 0.17). Taxonomic diversity directly affected productivity via CWM traits (indirect effect = −0.22) and structural complexity (indirect effect = 0.06). Topography directly affected taxonomic diversity (direct effect = 0.26), while having relatively minor impact on productivity (total effect = −0.04) (Fig 3).

## 4. Discussion

Tree species composition directly influences forest ecosystem functioning, with dominant species in a forest stand playing a particularly crucial roles. In this forest, the *Fagaceae* (specifically *C. henryi*) was the primary contributor to productivity (Fig 1a-b) and served as a dominant upper canopy species and a key community-building species in subtropical evergreen broad-leaved forests. Meanwhile, we observed that species productivity increased significantly with increasing species abundance (Fig 1c), indicating that common and companion species with higher abundances contribute more substantially to community productivity. Furthermore, plant functional traits influence productivity composition through

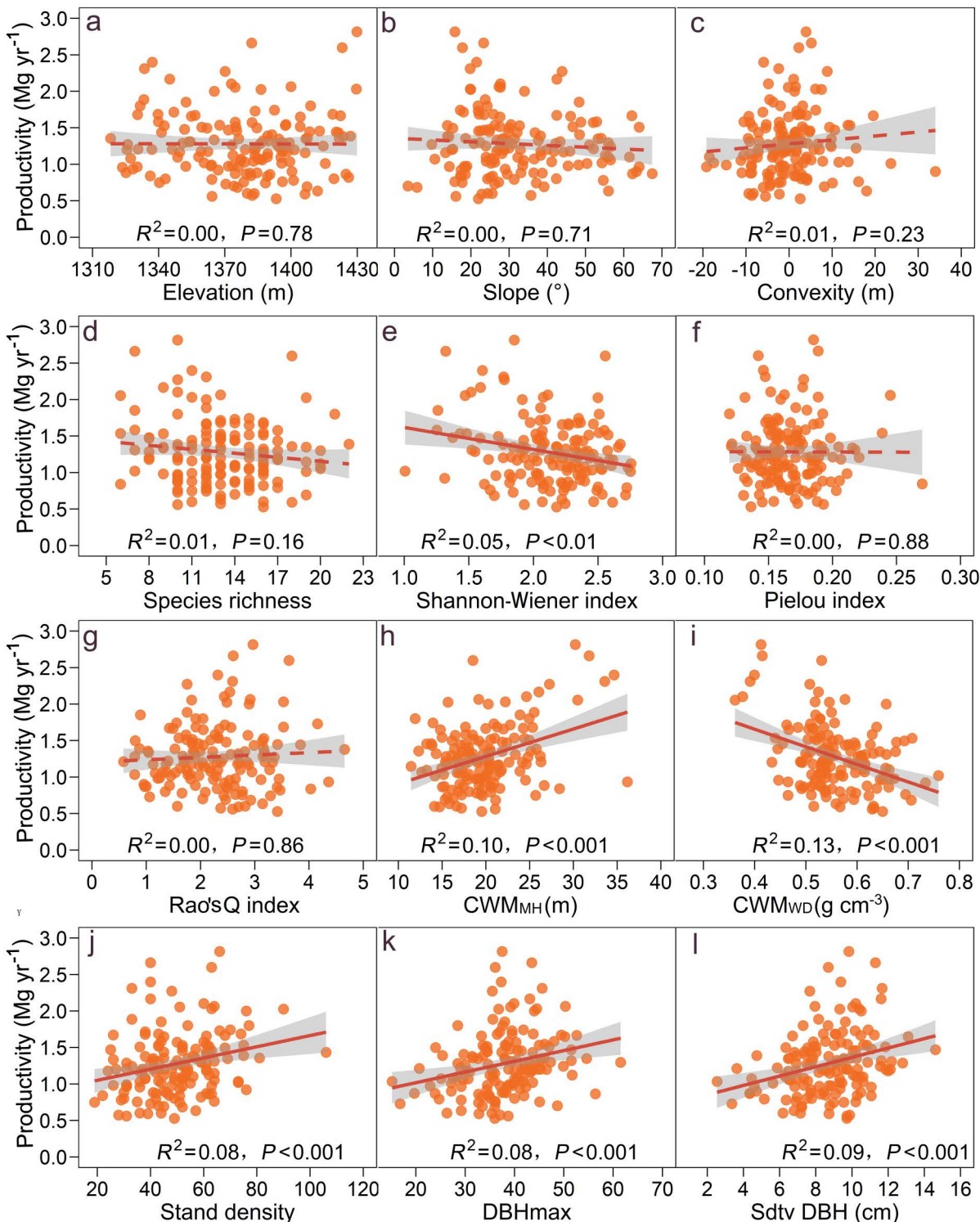

**Fig 2. Relationship between multiple predictor variables and forest productivity in 0.04-ha subplot level.** Filled areas indicate 95% confidence intervals. DBHmax, maximum DBH, Stdv DBH, standard deviation of DBH; $CWM_{WD}$, community-weighted mean of wood density; $CWM_{MH}$, community-weighted mean of maximum height.

**Table 1. Summary of the best multiple regression model for the effects of individual predictor variables on biomass and productivity.**

| Explantory variable | Estimate | SE | t-value | p-value |
|---|---|---|---|---|
| Intercept | −0.64 | 0.059 | 0.000 | 1.000 |
| Stand density | 0.542 | 0.067 | 8.108 | <0.001 |
| Stdv DBH | 0.306 | 0.067 | 4.571 | <0.001 |
| Rao's Q index | −0.139 | 0.069 | −2.011 | 0.0462 |
| $CWM_{MH}$ | 0.400 | 0.710 | 5.568 | <0.001 |
| $CWM_{WD}$ | −0.335 | 0.651 | −5.156 | <0.001 |

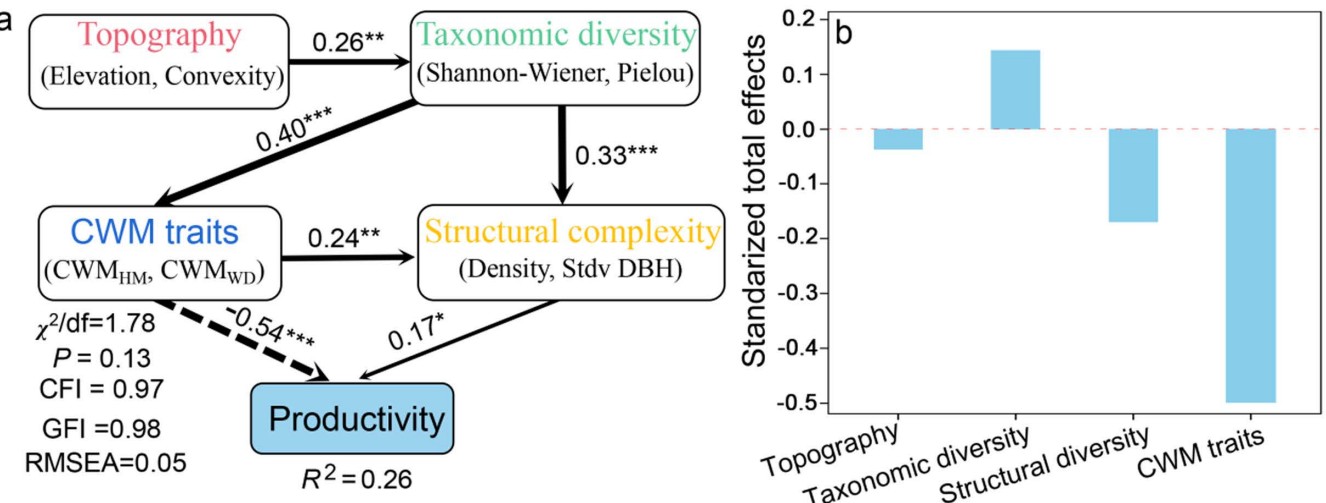

**Fig 3. Direct and indirect effects of multiple predictor factors on productivity.** The numbers on the arrows indicate standardized path coefficients. $R^2$ indicates the proportion of variance explained. *$P<0.05$; **$P<0.01$; ***$P<0.001$. Stdv DBH, standard deviation of DBH; $CWM_{WD}$, community-weighted mean of wood density; $CWM_{MH}$, community-weighted mean of maximum height.

species-specific difference in resource utilization. Clearly, tree maximum height was significantly and positively correlated with productivity at the species level (Fig 1d), while wood density was negatively correlated with productivity (Fig 1e).

Most existing studies suggest that taxonomic and functional diversity enhances forest productivity [33,6,58]. Our bivariate analysis indicated that three taxonomic indices and functional trait diversity (Rao's Q index) showed weak correlations with forest productivity, with all correlations being statistically non-significant except for the Shannon-Wiener index (Fig 2d-g). We propose two reasons to explain why our results contradict prior knowledge. On the one hand, at the 20×20 m subplot scale, most of the sampling sites had species richness ranging from 10 to 17. Most usually, the species richness effects tend to saturate at around 5–8, which may result in to missing out the positive effects of diversity. On the other hand, as species richness increases, the Rao's Q index tends to saturate, resulting in a non-significant correlation between function diversity and productivity. Collectively, we cannot exclude the potential effects of tree diversity.

Structural complexity plays a key role in determining wood productivity in forest ecosystems [31,59]. Stand structural attributes that reflect available resources, canopy complexity, intraspecific interactions of trees and niche differentiation and facilitation can potentially determine forest biomass and productivity [19,21,59,14]. In this natural forest ecosystem, our analysis showed that stand density, tree-size variability and maximum DBH were significantly and positively related to productivity (Fig 2j-i), consistent with previous studies in subtropical and temperate forests [31,33]. The beneficial impacts of these structural complexity variables on forest productivity indicate the presence of both complementarity and selection

effects in species-rich subtropical natural forests. Furthermore, when compared to taxonomic and functional diversity, structural diversity exerts a more pronounced influence on forest productivity [31,33] (Fig 3). Taxonomic and functional diversity indirectly enhance productivity by modulating variations in stand density and tree height variability (Fig 3), suggesting that alterations in taxonomic and functional diversity contribute to increased structural complexity within ecosystems, thereby positively influencing ecosystem functions [20,31,32]. This evidence further substantiates the existence of complementarity and selection effects within forest ecosystems.

Across various forest types and environment conditions, the traits of dominant tree species within a community serve as crucial determinants of forest carbon storage [60]. Furthermore, the direct influence of these trait values on ecosystem functioning represents a crucial component of the mass ratio hypothesis [10]. In our study, we employed two functional trait identities to reflect different aspects of species acquisition strategies: a stem trait (wood density) and a whole-plant trait (maximum height). Our results showed that $CWM_{MH}$ was significantly and positively related to productivity (Fig 2h), whereas $CWM_{WD}$ was significantly negatively correlated with forest productivity (Fig 2i). When other factors were accounted for in mixed models, $CWM_{MH}$ and $CWM_{WD}$ still had similar effects on forest productivity (Table 1). This finding emphasizes that resource-acquisitive species with greater maximum height and lower wood density contribute more to forest productivity than resource-conservative species with lower greater maximum height and high wood density [61], aligning with the mass-ratio hypothesis in these natural forests. In the SEM model, the impact of community-weighted mean (CWM) trait values on forest productivity is more pronounced than that of taxonomic diversity, functional diversity, and structural complexity; however, the interplay among these various factors collectively influences forest productivity (Fig 3). These findings indicate that ecosystem function in natural forest ecosystems is driven by multiple interrelated factors. The mass ratio hypothesis provides a more comprehensive explanation for the intrinsic driving forces behind forest ecosystem functions [14,45,60].

Local environmental changes can influence species distribution, abundance and plant life attributes, thereby indirectly or directly affecting forest biomass and productivity [14,35–37]. Topographic factors (e.g., elevation, slope and convexity) influence stand biomass and productivity via affecting tree diversity, stand structure, as well as water and light resource availability [35,36]. However, in this study, three topographic factors were not significantly correlated with productivity of evergreen deciduous broad-leaf mixed forests (Fig 2a-c). Given that elevation gradient of the sample plots ranged from 1310 to 1430 m, the minimal elevation difference resulted in an insignificant impact of elevation on productivity. Additionally, the slope and convexity associated with altitude variations did not significantly affect stand productivity. Our SEM analyses showed that topographic factors had the least direct effect on forest productivity (Fig 3b), while elevation and convexity directly caused changes in taxonomic diversity and indirectly regulated forest productivity (Fig 3a), but the direct and indirect effects of topographic factors on forest productivity were small (Fig 3). Collectively, our findings underscore the comparatively minor influence of habitat heterogeneity driven by topographic factors on ecosystem functionality.

## 5. Conclusions

This study evaluated the influence of topography, tree diversity, structural complexity, and functional trait identity on wood productivity in natural evergreen deciduous broad-leaf mixed forest systems. Our findings show that the habitat heterogeneity formed by topographic factors had only a weak effect on forest productivity. The effects of taxonomic and functional diversity on forest productivity demonstrate that the complementarity effect is not prominent. In contrast, the positive influences of structural diversity (i.e., stand density, large-diameter trees and tree-size variation) support the complementarity effect. This indicates that while taxonomic and functional diversity may not enhance productivity through complementarity, structural diversity plays a pivotal role in promoting forest productivity in this forest. Note that forest productivity was enhanced by traits with greater maximum height and lower wood density, thereby supporting the mass-ratio hypothesis. Our findings suggest that community-weighted mean traits are a primary driving force behind forest productivity, while the mass ratio hypothesis provides a fundamental mechanism for elucidating biodiversity and ecosystem functionality.

Ultimately, fostering natural structural complexity and fast-growing acquisitive species is beneficial for maintaining and enhancing ecosystem function in natural evergreen deciduous broad-leaf mixed forests.

## Author contributions

**Conceptualization:** Xiaoyu Chen, Meng Xiang, Qiang Zhang.

**Data curation:** Xiaoyu Chen, Meng Xiang, Lan Yao, Jiang Zhu, Qiuju Guo, Keyan Zhang.

**Formal analysis:** Xiaoyu Chen, Meng Xiang.

**Funding acquisition:** Qiang Zhang.

**Investigation:** Lan Yao, Xuru Ai, Qiuju Guo, Keyan Zhang.

**Methodology:** Xiaoyu Chen, Meng Xiang, Qiang Zhang.

**Project administration:** Qiang Zhang.

**Software:** Jiang Zhu.

**Supervision:** Xuru Ai, Qiang Zhang.

**Writing – original draft:** Xiaoyu Chen, Meng Xiang, Lan Yao, Xuru Ai, Qiuju Guo, Keyan Zhang.

**Writing – review & editing:** Xiaoyu Chen, Meng Xiang, Lan Yao, Xuru Ai, Jiang Zhu, Qiuju Guo, Keyan Zhang, Qiang Zhang.

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
