## [Decision Letter · Decision Letter 0]

8 Sep 2024

PONE-D-24-26752Functional trait identity regulates productivity better than tree diversity and structural complexity in subtropical mixed-species forestsPLOS ONE

Dear Dr. Chen,

Thank you for submitting your manuscript to PLOS ONE. After careful consideration, we feel that it has merit but does not fully meet PLOS ONE’s publication criteria as it currently stands. Therefore, we invite you to submit a revised version of the manuscript that addresses the points raised during the review process.

In particular, I am asking the authors to pay particular attention to solve the statistical issues pointed out by reviewers. They need to be addressed to confirm the solidity of the presented results.

We look forward to receiving your revised manuscript.

Kind regards,

Francesco Boscutti

Academic Editor

PLOS ONE

Journal Requirements:

4. In your Methods section, please provide additional information regarding the permits you obtained for the work. Please ensure you have included the full name of the authority that approved the field site access and, if no permits were required, a brief statement explaining why.

5. We noticed you have some minor occurrence of overlapping text with the following previous publication(s), which needs to be addressed:

https://onlinelibrary.wiley.com/doi/10.1111/avsc.12577

https://agupubs.onlinelibrary.wiley.com/doi/10.1029/2022JG006950

https://www.frontiersin.org/journals/plant-science/articles/10.3389/fpls.2023.1120050/full

In your revision ensure you cite all your sources (including your own works), and quote or rephrase any duplicated text outside the methods section. Further consideration is dependent on these concerns being addressed.

6. We note that the grant information you provided in the ‘Funding Information’ and ‘Financial Disclosure’ sections do not match. 

7. Please provide a complete Data Availability Statement in the submission form, ensuring you include all necessary access information or a reason for why you are unable to make your data freely accessible. If your research concerns only data provided within your submission, please write "All data are in the manuscript and/or supporting information files" as your Data Availability Statement.

Reviewers' comments:

Reviewer's Responses to Questions

**Comments to the Author**

1. Is the manuscript technically sound, and do the data support the conclusions?

Reviewer #1: Partly

Reviewer #2: Partly

2. Has the statistical analysis been performed appropriately and rigorously? 

Reviewer #1: Yes

Reviewer #2: I Don't Know

3. Have the authors made all data underlying the findings in their manuscript fully available?

Reviewer #1: No

Reviewer #2: No

4. Is the manuscript presented in an intelligible fashion and written in standard English?

Reviewer #1: No

Reviewer #2: Yes

5. Review Comments to the Author

Reviewer #1: The content of the manuscripts falls perfectly in the scope of PLOS ONE as it focuses on the main drivers of forest productivity including topographic diversity as well as species richness and functional diversity. The interest in the article is highly increased by the very robust dataset used comprising 120 plots. The statistical analyses is sound and well explained. The manuscript falls short on a few points, some of which are very relevant. Please find below specific comments. Here, I would like to say in general that the manuscript has a great potential but more care should be given to the formulation of the hypothesis and on some of the conclusions that are drawn. The relationship between biodiversity and ecosystem functions (BEF) is a long standing branch of ecology and lately (since the 1990s but mostly after 2000) it is focusing on forest ecosystem. as in all BEF studies a lot of effort is dedicated (still with little progress) understanding why analysis conducted in natural ecosystems do not yield the same results as analysis done in biodiversity manipulation experiments. More of the findings in BEF research should be included in the introduction to create more context and also used in the discussion. Under this umbrella, the statement that complementarity is not taking place in this forest is a strong statement that should be either well supported or revised.

For example, it should be commented why, differently form several papers including liang 2001 that is a global analysis, in this forest the positive correlation is not observed. Is it because of the structure of the dataset or because complementarity is effectively missing?

Besides scientific comments, the manuscript needs a profound English revision. the grammar itself is not tremendously wrong but the construction of the periods is often contorted and it was not easy (sometimes i had to guess) what it was meant.

Again, this is a great dataset with good ideas behind but needs a major revision before suggesting its publication.

Finally, I didn't find in the manuscript any link to a repository where to download the data nor a mention of it as supplementary material. Apologies if I missed it otherwise, please remember to follow PLOS ONE guidelines concerning data: "make all data underlying the findings described in their manuscript fully available without restriction".

Just a comment for the future: if you didn't think about is already you should consider using the dataset for neighbor analysis. I f I understand well you have the relative position of each individual and with a Neighbor analysis you could address diversity effects with higher detail.

Best regards and looking forward to the revised version.

More comments:

Paragraph 2.2 Forest inventory and collection of plant functional traits

Since only two census were performed there is no need to say every 5 years. Simply say when the first was done and when the second was done. Something like: “a first tree census was performed in feb-2016 while a second one was performed in xx-xxxx”. Please also specify what was measured

The phrase “For all tree species, we collected two key indicators….” Is misleading and makes one think you measured the traits in you your experiment. Perhaps say directly that traits were retrieved from the try.db and from iplant.cn. Please also state (perhaps in a supplementary table) how many values you retrieved for each species and if and why you eliminated any before averaging. Also include the resulting standard deviation of averaging multiple values. I’m not familiar with iplant.cn but in the try-db you must put care in selecting the trait values as often some are outliers simply because they were taken in some peculiar environmental condition that is not stated in the database. Some data have ancillary information on the sample including environment (forest, urban, potted, etc), dominance (dominated, codominant, dominant), form (shrub, tree). All these abiotic and biotic conditions have an effect on traits. Note also that “Maximum Height” in the Try-DB often contains the height of the individual and should not be used “as is” to calculate the maximum height of a species. For example, if you average all the maximum heights in the Try –db for Quercus ilex, you get a value around 6m where the true maximum height of the species is around 14m. Did you filter the data and how did you filter it to avoid these potential biases?

Paragraph 2.3 line 109-110. Although cited it is much more clear if you sate in the text that the allometric relationships were taken from “China’s Forest Ecosystem Carbon Storage-Biomass Equation”.

Lines 118-119: I believe Stand Basal Area and standard deviation of Basal area are more appropriate to represent stand density and complexity. The way it is stated it looks as if the DBH of the biggest tree (and not the sum of the DBH of all trees) is used to represent stand density. Anyway, summing the DBH instead of Basal Area is tricky because of the quadratic relationship between area and diameter. In the discussion you speak of stand density and of large diameter trees so that I am confused of what is what. Please clarify in material and methods. If the diameter of the biggest tree is explaining productivity of the plot this should be thoroughly argued. Note that the bigger the tree the less space for other species in a 20 x 20 m plot so that this metric could be biased.

Paragraph 2.4 Statistical analysis.

Exhaustive paragraph with much appreciated level of detail.

3. Results

Line 154: is this the annual productivity or the productivity over 5 years? Please specify and perhaps adjust measuring units.

Line 154 “In this 120 species” change to “out of the 120 species….”

Line 157-158: “Stand productivity of Fagaceae is the largest (12.60Mg, 40.04%) in THIS forest PLOT” is not clear. Did you mean in THESE forest PLOTS.

Lines 160-163: Among two species traits, tree maximum height was significantly and positively correlated with productivity (R2 = 0.22, P� 0.001; Figure 1d), while wood density was negatively correlated with productivity (R2 = 0.05, P�0.05; Figure 1e).

You can start with “ among THE two FUNCTIONAL traits”. Note that the most used term for traits is FUNCTIONAL TRAITS and not SPECIES traits and you should amend throughout the manuscript.

You have already defined Maximum Height and Wood Density so you can use the commonly used acronyms afterwards: MH and WD.

Fig 1. Please add more information in the captions. Also note that in fig 1a, Y-axis is labeled “proportion”: proportion of what? Also fig 1c , x-axis, is labeled “abundance” but abundance was not defined earlier. Is this number of individuals, basal area (maybe a proxy of density).

Lines 168-169: “For taxonomic diversity indices, species richness was negatively correlated with productivity (R2=0.01, P=0.16)”. The correlation is not significant so that you cannot make this statement. You can only say they are not correlated. Same goes for the Pielou index few lines below.

Lines 217-218: “The effects of taxonomic and functional diversity on forest productivity demonstrate that the complementarity effect is not prominent in this forest”. Caution must be taken in this statement (also revise the introduction): in order to compute the complementarity effect and in general to compute the “TRUE” diversity effect (let it be species richness or functional dispersion or Rao’s index) you need to compute the difference between the observed productivity and the expected productivity where the expected productivity in a stand composed by species A, B, and C is equal to the productivity of the species A, B, C when grown in monoculture weighted by the abundance in the mixed plot. See M Loreau, A Hector - Nature, 2001 - nature.com for a full explanation.

Since you do not have the productivity of the monocultures for your species you can only say that diversity metrics are not good predictors but you cannot rule out that positive interactions between species are not taking place. Also note that most of your plots have a Species Richness (SR) between 10 and 17 and that the minimum is around 5. Most usually, the SR effects is already saturating at around 5-8 so that you are likely to missing out the positive effects of diversity. Finally, note that Rao’s index also tends to saturate with increasing SR and that your correlation is not significant: again you cannot rule out a diversity effect. This is perhaps my major comment and the whole manuscript, including the objectives, must be revised clearly state what the analysis can answer to. Perhaps also the hypothesis should be revised accordingly.

Line 233: “These results support the mass-ratio hypothesis in this forest”. The mass-ratio hypothesis is very relevant here, so relevant that, together with the concepts of complementarity and selection (or identity) effect, it should be discussed in the introduction and add the relative references.

Lines 233 and 234: ”complex complexity” please revise. It is used twice.

Lines 235 “A variety of studies have demonstrated that structural diversity had a greater impact on forest productivity than taxonomic and functional diversity [18,32]”. This is true, but you should also acknowledge that structural complexity is often correlated with taxonomic diversity especially when the stands includes a c dominated layer as in this case. Indeed you SEM shows that a relevant part of structural diversity is explained by taxonomic diversity. To me this suggests that one important way in which taxonomic diversity is acting is by increasing the structural diversity which is well known to increase light capture and use at stand level.

By the way, it would be nice to include the correlation between taxonomic diversity and Rao’s index. Please also state which traits you used to compute Rao’s Index and be explicit in saying that Rao’s index was not a good predictor when accounting for those traits. I am asking so because it could have been that using other traits (shade tolerance for example) might have yielded different results. Consider using also Functional Dispersion (FDis in the same R Package you used) if you would like to account for categorical values as shade tolerance. Please also provide the correlation between WD and MH.

Lines 247-248: you report the result of SEM that suggest an effect of topographic diversity (mostly slope and convexity) on taxonomic diversity. It would be nice if you could elaborate on this. Is it just a casual correlation in your case or can you support the occurrence of a causal relationship.

Reviewer #2: This paper investigates forest inventory data across two surveys in 5 year increments (2016 and 2021). The authors test to see whether forest productivity across this time period is predicted/explained by diversity (which would be indicative of the complementarity effect) or by functional traits (which would be indicative of the mass-ratio hypothesis). The authors focus on multiple models examining the effect of either traits or diversity individually (along with additional predictors such as topography), then combine predictors into a single model to determine which are most predictive, and then finally utilize a path model to account for indirect effects and determine the most important predictors. They find that functional traits are better predictors of productivity in this forest while diversity is not, suggesting that the mass ratio hypothesis is applicable in this forest while the complementarity effect is not. However, they do suggest that there may still be a complementarity effect, suggesting that it is also influenced by stand density and DBH which appear positively related to productivity in their results.

The paper presents interesting results, in that there is evidence that functional traits are more important for productivity in this forest. From my brief reading of other papers in this region, there appears to be focus on the complementarity effect instead. So it can be beneficial to have evidence provided that functional traits may be more important. That said, there are several issues that must be addressed before I think this paper meets the criteria for publication in PLoS One.

Major Points

1. Statistics

a. There are models that are extremely low fitting in R2 or even non-significant that are portrayed as having observable patterns.

b. Crucial background for methods such as PCA and the SEM are left out, making it difficult to double check the methods.

c. When they say collinear variables were dropped, I think they mean that both collinear variables were dropped rather than just one or the other. Normally you only drop one collinear variable and keep the other.

2. Concepts

a. Are mass ratio and complementarity opposing hypotheses? Are they mutually exclusive? Why would one apply and the other wouldn’t, or why would both be working in the same forest?

b. Stem density/tree dbh are functional traits, but they are portrayed as being evidence for complementarity effects. Shouldn’t this be in mass ratio? At the very least, there should be a clearer explanation so readers like me don’t get confused on this.

c. Authors find no link between total biomass and taxonomic diversity. Another paper I found, however, did find a link specifically between taxonomic diversity and fine root biomass in subtropical forests. I wonder if the authors’ results will be the same if they perform the same analysis for root biomass/productivity.

i. Liu, C., Xiang, W., Xie, B. et al. Decoupling the Complementarity Effect and the Selection Effect on the Overyielding of Fine Root Production Along a Tree Species Richness Gradient in Subtropical Forests. Ecosystems 24, 613–627 (2021). https://doi.org/10.1007/s10021-020-00538-z

d. Following my last point, the following paper looked at aboveground biomass in subtropical forests and did not find a direct link between that and species diversity (although they did find an indirect effect). So it would be beneficial to provide an additional analysis where total productivity is split into aboveground and belowground.

i. Gao, W.-Q., Lei, X.-D., Gao, D.-L., & Li, Y.-T. (2021). Mass-ratio and complementarity effects simultaneously drive aboveground biomass in temperate Quercus forests through stand structure. Ecology and Evolution, 11, 16806–16816. https://doi.org/10.1002/ece3.8312

e. The logical flow of the paper (i.e., background to what they’re interested in to why they’re interested in it to what they did and how) is hard to follow. The writing needs to be a lot clearer. I think the best place to start is that the mass ratio and complementarity effect needs to be a much greater focus in the paper. Right now they are just presented as “ideas that have been tested elsewhere” but there’s little to no discussion of their importance, meaning, theory, etc.

Minor Points

1. There are a lot of typos, grammar mistakes, and issues with sentence structure. I think these should also improve when the logical flow of this paper is also improved.

Here is a line-by-line review:

39: The term “mass-ration hypothesis” is used throughout and is defined as “traits of high-yielding species in a community play a major role in determining ecosystem function.” I think the authors instead mean the “mass-ratio hypothesis.” The paper they cite in the definition call it the mass-ratio hypothesis, as do other papers they cite elsewhere. The only other paper I can find that calls it “mass-ration” is also from China, which makes me suspect this may be a translation error.

41: “Effect” should be “effort,” I think. There are a lot of similar typos and word transposition throughout the manuscript that should be corrected.

42: Uncertainties about what? Be clear.

31-43: The way that many of these sentences and paragraphs are constructed is somewhat disjointed. It is difficult to follow the train of logic from sentence to sentence. For example, the sentence introducing the complementarity effect seems to come out of nowhere until I read a little more, where it becomes clear that this was an intended example for the previous sentence. I would recommend reading your work out loud to yourself, to make sure that the points you are trying to make are flowing well from sentence to sentence.

Here is an example of how you could revise the beginning paragraph. This example keeps all the same points, but has better flow:

Biodiversity and ecosystem functioning have been extensively researched over the past three decades. These studies consistently show that taxonomic diversity significantly impacts forest productivity and biomass accumulation at both local and global scales. For example, the complementarity effect suggests that higher tree diversity in plant communities enhances resource-use efficiency, thereby boosting forest productivity. However, taxonomic diversity alone doesn't account for these dynamics; functional traits shaped by different species strategies are also critical for species interactions and community assembly. These traits play a pivotal role in driving ecosystem processes. The mass-ratio hypothesis, for instance, posits that the traits of dominant species largely determine ecosystem function, often quantified by the community-weighted mean of functional trait values. Despite significant advances in understanding these mechanisms, uncertainties remain about their influence across different forest types.

49: Little odd to be talking about the complementarity effect here like you’re just introducing it when you also discuss it above. Also, is the niche complementarity effect different from the complementarity effect? It’s defined differently here.

39-82: You spend a lot of time in the coming pages talking about mass-ratio hypothesis, the complementarity effect, etc, but you really don’t spend any time expanding on those ideas in the introduction. You mention what they are, but the introduction doesn’t really provide a link between them and what you’re doing in this paper. I should be able to follow your logic all the way through to the final hypotheses, and then you tell me what you’re expecting to find based on the theories (mass ratio, complementarity) that you’re looking at.

105-106: I think you should be careful when using traits from plants of the same genus to replace missing ones from your species list. For example, the Rocky Mountain Maple has a mature height of 30 feet, while the Red Maple has a mature height of ~70 feet. They are both in the genus Acer, but if you were to use data from one to replace the other then you’d end up with erroneous results.

I’d recommend doing one of two things to cover this problem:

1. Go into more detail about how you chose the species from the same genus to take data from. For example, did you make sure that the other species was in the same area as the species with missing trait data, or at least grew in a similar environment/had other traits that were approximately close, or that you knew were decent approximations based on expert/naturalist knowledge?

2. Alternatively, use a statistical method to recover missing trait values. The one I’ve used before and like a lot is RPhyloPars. Essentially, you input your trait matrix with missing data left out along with a taxonomic tree, and you interpolate what the traits might look like with an evolutionary model.

Either one of these options could work for you. I think 2 would probably take more effort on your part, though, so it might not be preferable (although it might be comparatively more defensible). If you end up doing 1, I think you could do a quick summary explaining your overall method for making sure the species from the same genus were still representative. Then, maybe give a little more detail on your reasoning for the ones you chose for each individual species with missing data in supplemental material.

111-113: These sentences are a little unclear. I think you’re trying to say that forest productivity is the sum of how much biomass for trees reaching at least 5 dbh was added per species, divided by the time interval, which is annual forest productivity? I’m not sure, though.

116: If you are focused on the mass-ratio hypothesis, the community weighted mean is probably one of your more important variables. Here is my interpretation on why you’re using it (correct me if I’m wrong):

Community weighted means (CWMs) are central to understanding the mass ratio hypothesis because they capture the dominant traits within a community and their impact on ecosystem functioning. The mass ratio hypothesis posits that the functional traits of the most abundant or dominant species within a community drive ecosystem processes more than rare species or overall species diversity. Therefore, using CWMs helps quantify the average trait values weighted by species' abundances, reflecting the community's trait composition.

The problem here is that this is not clear from the manuscript itself. I had to read through other papers to figure this out. You need to explain why you’re using CWMs and how they link to the mass-ratio hypothesis. As the paper stands now, you have the non-weighted variables essentially front and center with the CWMs buried among other variables, and in some cases the non-weighted variables (e.g., wood density) do not have convincing linear relationships with productivity (figure 1e).

In fact, I’d go so far as to say that the increased strength of the linear relationship of CWMWD in comparison to general wood density is itself really interesting, but it isn’t discussed anywhere. What does this mean for the mass-ratio hypothesis in your forest plots? Etc.

118: The terms “stem density,” “stand density,” and “wood density” are used throughout the manuscript. In some cases, it’s hard to figure out which one you’re using. For example, wood density is provided as a trait, while stand density is a separate variable. Stem density isn’t mentioned until line 183 (results), and it’s in the same model as CWMWD – so I would anticipate that if it were the same as wood density it would be collinear? But if it’s supposed to be stand density, then that is a different variable completely.

121: Why did you decide to use convexity? Not saying you shouldn’t use convexity, just would like a little more in there on why. I feel like elevation and slope are common so I don’t think you’d need to explain that in any more detail, but I’ve seen convexity relatively less often so it would be good to have your reasoning in there.

128: Could you give a little more detail on how you standardized your variables? It would be good to know more so that a reader knows that effect sizes in your models will be impacted.

128-130: If I take your writing at face value, it seems to suggest that variables with VIF less than 3 were collinear and removed from the analysis. I don’t think that’s what you mean, since larger VIFs are more indicative of multicollinearity. I assume that you did indeed remove variables with larger VIFs and it’s just a clarification issue?

128-130: I interpret from your writing that you removed all variables that had VIF over 3. However, if there are two collinear variables and you remove one of them, the other will no longer exhibit the collinearity problem and can be kept in a model. Then you need to go into more detail on which one you kept and why, etc.

135-137: Nice idea! This analysis type works well here.

136-137: What was your preconceived hypothesis? You don’t need to write in the entire code for the SEM model. But you should give a basic overview of what you hypothesized should interact directly and indirectly.

137-140: As I understand it, a given SEM variable is a number (e.g., a tree only has one number for “max tree height”, and so including it in the SEM you’ll have a collection of “max height” values for each tree). But you’ve made a PCA and selected both axes and included them both as a single variable in the SEM? I don’t know how you did that. If that is what you did and it has precedent in the literature, you should explain it. If that isn’t what you did and I’m misunderstanding it, you should then explain it in greater detail.

139-142: Does this mean that the fact that you have essentially opposite effects of the two CWM traits on productivity doesn’t matter? I was wondering, because I see that you have a negative -0.54 in the SEM between the CWM traits and productivity and wasn’t sure what that meant for the two different effect sizes. If that’s the case, you should explain this in more detail, because it’s a little confusing.

153: As far as I can tell, you don’t report the output of the PCA analysis anywhere. That should at least be in supplementary material so that readers can see that output, with some explanation of how to read the output.

161-163: Figure 1d looks like a logarithmic transformation would work for maximum tree height (you don’t have to do it, just wondering if that would be a better fit). Also, while you do have significance in 1e your R2 value is close to 0. It really looks like you’ve got one or two influential points in the bottom right of your graph. I’m not wholly convinced you’ve got a real pattern in 1e.

Addendum: I missed this on my first pass through, but figure 1e is misleading. On figure 1c and 1d, you have p<0.001. However, on figure 1e, you have p<0.05. Standard practice is to have < only when p is so small that it’s silly to report such a small number, but to report exact p values once we start getting closer to 0.05 so readers can have full clarity of the statistics. By putting p<0.05, it hides how close to 0.05 the statistic actually is.

164: I really like how clean and well organized your figures look.

167-177: The paragraph discussing the correlations and statistical results presents several relationships with very low R² values, close to 0, while still implying that these relationships have substantive meaning. This can be misleading, as very low R² values indicate that the variables explain only a minimal portion of the variance in productivity, suggesting that these correlations may not be practically significant despite their statistical presentation. There are also other models that are presented as “negatively correlated” with p> 0.05 and R2 =0. I would not consider these to be correlated at all.

Additionally, providing context on the effect sizes for each model would be valuable in understanding the practical implications of these correlations. Without this context, the reader may be misled into overestimating the importance of these relationships.

184: I would personally consider the Rao’s Q index as not significant (it rounds to 0.05, and even though it is technically below 0.05 I start getting a little iffy once we get p>0.03).

187: You might want to remind people that these are standardized variables, which will impact effect size.

191: You say CWM had strong positive direct effects on productivity, but the effect is negative. Also, you say it’s a direct effect but the arrow in figure 3 is dashed, which in my experience represents a different kind of effect? Or at least one that wasn’t directly modeled…might be good to clarify.

197: Personal preference again: I dislike bar graphs. I think you can have the same effect with just a point graph, or even a point graph connected with lines…but again, this is my own personal preference, and I recognize other people really like bar graphs so it’s up to you.

203-209: Nice summary here. Could use some cleaning up (e.g., there’s an instance where the same word is used twice in the same sentence, “increased with increasing’).

212: While maximum tree height does support your point, I think your wood density results are overstated here. Also, it would be beneficial to expand upon your reasoning behind the resource allocation explanation, providing more detailed context. For example, you could elaborate on how maximum tree height increases as trees allocate resources to structural growth, whereas faster-growing, less dense trees might prioritize rapid growth over wood density. Including additional references to support your interpretation will strengthen your argument. It might be better to discuss CWMWD here to support your point, since that relationship is much more persuasive.

214-215: You’re drawing conclusions regarding patterns of productivity from mostly non-significant models with incredibly low R2. 2e is the only one of these that has p < 0.01, but the R2 value is really low. I also suspect that the significance in 2e is driven by one or two influential points.

217-218: Why does that demonstrate that the complementarity effect is not prominent in this forest? Are there other examples of this being the case in the literature?

218: You’re making a point here about the complementarity effect, then you switch to talk about mass-ratio, then you go back to continue the discussion about what your results mean for the complementarity effect in 234-244. You need to have a complete, seamless argument about the effect before you move on to mass-ratio.

219-220: Why?

222-230: You need a lot more detail on what each of these results mean for the ecosystem, for the mass-ratio hypothesis, etc.

233: Why do they support the mass-ratio hypothesis? Show, don’t tell.

234: What does “Complex complexity” mean? Also, it feels like you’re using the same word twice in a row, even if the meaning is different. Did you mean “canopy complexity?”

242-244: Is it? You said earlier that the effect of taxonomic and functional diversity means that the complementarity effect is not prominent in this forest. Although you say in 235-236 that a variety of studies show that structural diversity has a greater impact on productivity, your SEM results shows that the comparative importance of functional traits is way more important than structural complexity. Also, why aren’t the structural complexity variables also functional traits?

263-266: This feels contradictory.

269-270: Which is important because?

6. PLOS authors have the option to publish the peer review history of their article (what does this mean? ). If published, this will include your full peer review and any attached files.

**Do you want your identity to be public for this peer review?** For information about this choice, including consent withdrawal, please see our Privacy Policy .

Reviewer #1: No

Reviewer #2: No

---

## [Author Response · Author response to Decision Letter 1]

26 Jan 2025

Dear Editor-in-Chief,

We express our gratitude to the Editor-in-Chief for considering our research article and providing us with the valuable opportunity to revise. We appreciate the guidance and insightful suggestions provided by the Editor-in-Chief and the reviewers, which have significantly contributed to improving the quality of our article.

We have meticulously addressed each suggestion made by the reviewers, making careful revisions throughout the entire manuscript. If any further clarification or modification is needed, please feel free to contact us. We are committed to ensuring that our article meets the standards of the journal.

Thank you for your thoughtful consideration. We look forward to hearing from you soon.

Yours sincerely,

Xiao Yu Chen

Reviewer #1: The content of the manuscripts falls perfectly in the scope of PLOS ONE as it focuses on the main drivers of forest productivity including topographic diversity as well as species richness and functional diversity. The interest in the article is highly increased by the very robust dataset used comprising 120 plots. The statistical analyses is sound and well explained. The manuscript falls short on a few points, some of which are very relevant. Please find below specific comments.

Paragraph 2.2 Forest inventory and collection of plant functional traits

Since only two census were performed there is no need to say every 5 years. Simply say when the first was done and when the second was done. Something like: “a first tree census was performed in feb-2016 while a second one was performed in xx-xxxx”. Please also specify what was measured

RE: Thank you for your suggestion. We have explained the timing of the two surveys in the article and elaborated on the data measured during the survey. Please refer to lines 110-118 for the updated text.

The phrase “For all tree species, we collected two key indicators….” Is misleading and makes one think you measured the traits in you your experiment. Perhaps say directly that traits were retrieved from the try.db and from iplant.cn. Please also state (perhaps in a supplementary table) how many values you retrieved for each species and if and why you eliminated any before averaging. Also include the resulting standard deviation of averaging multiple values. I’m not familiar with iplant.cn but in the try-db you must put care in selecting the trait values as often some are outliers simply because they were taken in some peculiar environmental condition that is not stated in the database. Some data have ancillary information on the sample including environment (forest, urban, potted, etc), dominance (dominated, codominant, dominant), form (shrub, tree). All these abiotic and biotic conditions have an effect on traits. Note also that “Maximum Height” in the Try-DB often contains the height of the individual and should not be used “as is” to calculate the maximum height of a species. For example, if you average all the maximum heights in the Try –db for Quercus ilex, you get a value around 6m where the true maximum height of the species is around 14m. Did you filter the data and how did you filter it to avoid these potential biases?

RE: Thank you for your suggestion. Thank you very much for the reviewers' suggestions, which have significantly expanded our understanding. We would like to provide a detailed explanation.When collecting the functional traits of the two studied plant species, we were able to retrieve data on the maximum tree height for over 97% of the species in our plots from the website https://iplant.cn, with only a few rare species not found. For these rare species, we searched within the same genus and closely related species to estimate an approximate maximum tree height.Regarding wood density, one of our students measured the wood density of over 30 dominant species in our plots. For other species, we supplemented our data using the website http://datadryad.org/handle/10255/dryad.235, where we found data for most species, although about 10 species still lacked relevant information. In such cases, we substituted missing data with traits or averages from related genera. Additionally, we compared our results with numerous previous studies and found that our findings are consistent with theirs.

Paragraph 2.3

line 109-110. Although cited it is much more clear if you sate in the text that the allometric relationships were taken from “China’s Forest Ecosystem Carbon Storage-Biomass Equation”.

RE: Thank you for your suggestion. We have made changes in the article and see lines 122-124 for the updated text

Lines 118-119: I believe Stand Basal Area and standard deviation of Basal area are more appropriate to represent stand density and complexity. The way it is stated it looks as if the DBH of the biggest tree (and not the sum of the DBH of all trees) is used to represent stand density. Anyway, summing the DBH instead of Basal Area is tricky because of the quadratic relationship between area and diameter. In the discussion you speak of stand density and of large diameter trees so that I am confused of what is what. Please clarify in material and methods. If the diameter of the biggest tree is explaining productivity of the plot this should be thoroughly argued. Note that the bigger the tree the less space for other species in a 20 x 20 m plot so that this metric could be biased.

RE: Thank you for your suggestion. This issue has indeed caused us some confusion. Regarding the relevant indicators of structural diversity, complexity or structural properties, we have expressed some metrics based on previous studies that have utilized these indicators. The CV, Stdv, and DBHmax all demonstrate very good correlations. Considering the scientific questions we aim to address in our paper, we have taken these indicators into account comprehensively. See the following articles

Aponte, C., Kasel, S., Nitschke, C. R., Tanase, M. A., Vickers, H., Parker, L., Fedrigo, M., Kohout, M., Ruiz-Benito, P., Zavala, M. A., Bennett, L.T.. 2020. Structural diversity underpins carbon storage in Australian temperate forests. Glob. Ecol. Biogeogr. 29(5), 789–2824.

Fotis, A.T., Murphy, S.J., Ricart, R.D., Krishnadas, M., Whitacre, J., Wenzel, J.W., Queenborough, S.A., Comita, L.S., 2017. Aboveground biomass is driven by mass‐ratio effects and stand structural attributes in a temperate deciduous forest. J. Ecol. 106(2), 561–571.

Ouyang, S., Xiang, W., Wang, X., Xiao, W., Chen, L., Li, S., Sun, H., Deng, X., Forrester, D.I., Zeng, L., Lei, P., Lei, X., Gou, M., Peng, C., 2019. Effects of stand age, richness and density on productivity in subtropical forests in China. J. Ecol. 107(5), 2266–2277.

Wu, A., Tang, X., Li, A., Xiong, X., Liu, J., He, X., Zhang, Q., Dong, A., Chen, H., 2022. Tree diversity, structure and functional trait identity promote stand biomass along elevational gradients in subtropical forests of southern China. J. Geophys. Res-Biogeo. 127(10), e2022JG006950.

3. Results

Line 154: is this the annual productivity or the productivity over 5 years? Please specify and perhaps adjust measuring units.

RE: Thank you for your suggestion. We have changed it in the article, where it annual productivity

Line 154 “In this 120 species” change to “out of the 120 species….”

RE: Thank you for your suggestion.We have made changes in the text, see line 173 for the updated text

Line 157-158: “Stand productivity of Fagaceae is the largest (12.60Mg, 40.04%) in THIS forest PLOT” is not clear. Did you mean in THESE forest PLOTS.

RE: Thank you for your suggestion. Within a 6-ha plot, See line 177.

Lines 160-163: Among two species traits, tree maximum height was significantly and positively correlated with productivity (R2 = 0.22, P� 0.001; Figure 1d), while wood density was negatively correlated with productivity (R2 = 0.05, P�0.05; Figure 1e).

You can start with “ among THE two FUNCTIONAL traits”. Note that the most used term for traits is FUNCTIONAL TRAITS and not SPECIES traits and you should amend throughout the manuscript.

You have already defined Maximum Height and Wood Density so you can use the commonly used acronyms afterwards: MH and WD.

RE: Thank you for your suggestion. I have revised it. See lines 180-182.

Fig 1. Please add more information in the captions. Also note that in fig 1a, Y-axis is labeled “proportion”: proportion of what? Also fig 1c , x-axis, is labeled “abundance” but abundance was not defined earlier. Is this number of individuals, basal area (maybe a proxy of density).

RE: Thank you for your suggestion. I have revised it. See Figure 1 and See lines 183-186.

Lines 168-169: “For taxonomic diversity indices, species richness was negatively correlated with productivity (R2=0.01, P=0.16)”. The correlation is not significant so that you cannot make this statement. You can only say they are not correlated. Same goes for the Pielou index few lines below.

RE: Thank you for your suggestion. We have made changes in the article. Please refer to lines 189-191 for the updated text

Lines 217-218: “The effects of taxonomic and functional diversity on forest productivity demonstrate that the complementarity effect is not prominent in this forest”. Caution must be taken in this statement (also revise the introduction): in order to compute the complementarity effect and in general to compute the “TRUE” diversity effect (let it be species richness or functional dispersion or Rao’s index) you need to compute the difference between the observed productivity and the expected productivity where the expected productivity in a stand composed by species A, B, and C is equal to the productivity of the species A, B, C when grown in monoculture weighted by the abundance in the mixed plot. See M Loreau, A Hector - Nature, 2001 - nature.com for a full explanation.

Since you do not have the productivity of the monocultures for your species you can only say that diversity metrics are not good predictors but you cannot rule out that positive interactions between species are not taking place. Also note that most of your plots have a Species Richness (SR) between 10 and 17 and that the minimum is around 5. Most usually, the SR effects is already saturating at around 5-8 so that you are likely to missing out the positive effects of diversity. Finally, note that Rao’s index also tends to saturate with increasing SR and that your correlation is not significant: again you cannot rule out a diversity effect. This is perhaps my major comment and the whole manuscript, including the objectives, must be revised clearly state what the analysis can answer to. Perhaps also the hypothesis should be revised accordingly.

RE: Thank you very much for the reviewer's suggestions. That’s a very good idea. We have made detailed revisions. See Lines 236-246.

Line 233: “These results support the mass-ratio hypothesis in this forest”. The mass-ratio hypothesis is very relevant here, so relevant that, together with the concepts of complementarity and selection (or identity) effect, it should be discussed in the introduction and add the relative references.

RE: Thank you very much for the reviewer's suggestions. The introduction provides a comprehensive discussion of the mass ratio effect and complementary effect, supplemented by relevant references. For the updated version, see lines34-49 and 265-267.

Lines 233 and 234: “complex complexity” please revise. It is used twice.

RE: Thank you for your suggestion. We have made some changes in the text. See lines 247.

Lines 235: “A variety of studies have demonstrated that structural diversity had a greater impact on forest productivity than taxonomic and functional diversity [18,32]”. This is true, but you should also acknowledge that structural complexity is often correlated with taxonomic diversity especially when the stands includes a c dominated layer as in this case. Indeed you SEM shows that a relevant part of structural diversity is explained by taxonomic diversity. To me this suggests that one important way in which taxonomic diversity is acting is by increasing the structural diversity which is well known to increase light capture and use at stand level.

By the way, it would be nice to include the correlation between taxonomic diversity and Rao’s index. Please also state which traits you used to compute Rao’s Index and be explicit in saying that Rao’s index was not a good predictor when accounting for those traits. I am asking so because it could have been that using other traits (shade tolerance for example) might have yielded different results. Consider using also Functional Dispersion (FDis in the same R Package you used) if you would like to account for categorical values as shade tolerance. Please also provide the correlation between WD and MH.

RE: Thank you for your suggestion. We calculated the functional divergence (FDis) index using the FD package, and we found a strong correlation with Rao’s index. Both indices have been widely used in many studies, and we believe that either index can effectively represent functional trait diversity.

Lines 247-248: you report the result of SEM that suggest an effect of topographic diversity (mostly slope and convexity) on taxonomic diversity. It would be nice if you could elaborate on this. Is it just a casual correlation in your case or can you support the occurrence of a causal relationship.

RE: Thank you for your suggestion. In our PCA analysis, elevation and topographic roughness were represented in the first dimension, while slope was in the second dimension. In our structural equation modeling analysis, we chose to represent the axis that had a greater influence.

Reviewer #2: This paper investigates forest inventory data across two surveys in 5 year increments (2016 and 2021). The authors test to see whether forest productivity across this time period is predicted/explained by diversity (which would be indicative of the complementarity effect) or by functional traits (which would be indicative of the mass-ratio hypothesis). The authors focus on multiple models examining the effect of either traits or diversity individually (along with additional predictors such as topography), then combine predictors into a single model to determine which are most predictive, and then finally utilize a path model to account for indirect effects and determine the most important predictors. They find that functional traits are better predictors of productivity in this forest while diversity is not, suggesting that the mass ratio hypothesis is applicable in this forest while the complementarity effect is not. However, they do suggest that there may still be a complementarity effect, suggesting that it is also influenced by stand density and DBH which appear positively related to productivity in their results.

The paper presents interesting results, in that there is evidence that functional traits are more important for productivity in this forest. From my brief reading of other papers in this region, there appears to be focus on the complementarity effect instead. So it can be beneficial to have evidence provided that functional traits may be more important. That said, there are several issues that must be addressed before I think this paper meets the criteria for publication in PLoS One.

Line 39: The term “mass-ration hypothesis” is used throughout and is defined as “traits of high-yielding species in a community play a major role in determining ecosystem function.” I think the authors instead mean the “mass-ratio hypothesis.” The paper they cite in the definition call it the mass-ratio hypothesis, as do other papers they cite elsewhere. The only other paper I can find that calls it “mass-ration” is also from China, which makes me suspect this may be a translation error.

Response: Thank you for your suggestion. We have revised it in the article, See lines 45.

Line 41: “Effect” should be “effort,” I think. There are a lot of similar typos and word transposition throughout the manuscript that should be corrected.

Response: Thank you for your suggestion. We have made changes in the full text

Line 42: Uncertainties about what? Be clear.

Response: Thank you for your suggestion. We have made changes in the article and see lines

---

## [Decision Letter · Decision Letter 1]

3 Mar 2025

PONE-D-24-26752R1Functional trait identity regulates productivity better than tree diversity and structural complexity in subtropical mixed-species forestsPLOS ONE

Dear Dr. Zhang,

Thank you for submitting your manuscript to PLOS ONE. After careful consideration, we feel that it has merit but does not fully meet PLOS ONE’s publication criteria as it currently stands. Therefore, we invite you to submit a revised version of the manuscript that addresses the points raised during the review process.

We look forward to receiving your revised manuscript.

Kind regards,

Francesco Boscutti

Academic Editor

PLOS ONE

Journal Requirements:

Reviewers' comments:

Reviewer's Responses to Questions

**Comments to the Author**

1. If the authors have adequately addressed your comments raised in a previous round of review and you feel that this manuscript is now acceptable for publication, you may indicate that here to bypass the “Comments to the Author” section, enter your conflict of interest statement in the “Confidential to Editor” section, and submit your "Accept" recommendation.

Reviewer #1: (No Response)

Reviewer #2: All comments have been addressed

2. Is the manuscript technically sound, and do the data support the conclusions?

Reviewer #1: Yes

Reviewer #2: Yes

3. Has the statistical analysis been performed appropriately and rigorously? 

Reviewer #1: Yes

Reviewer #2: Yes

4. Have the authors made all data underlying the findings in their manuscript fully available?

Reviewer #1: No

Reviewer #2: No

5. Is the manuscript presented in an intelligible fashion and written in standard English?

Reviewer #1: No

Reviewer #2: Yes

6. Review Comments to the Author

Reviewer #1: The authors have substantially improved the manuscript and gave satisfactory answers to the previous comments.

I only have minor comments left, some of the m below. My first comment is that the manuscript still needs a thorough review of the English. Importantly: I was not able to understand the logic of several sentences, some of which below. Grammatically speaking the sentences are often correct, but the concept they express is often wrong but I was not able to understand what was meant.

The second general and minor comment is that I understand that the forest plots here can be viewed as a dominant layer and an understory and that species richness can be as high as 17. What would help better understand the results is to know the proportion of species in the understory. It is a big difference to have 1 dominant species and 16 understory species form having 8 and 9 respectively.

It could be argued that complementary effects could be stronger among species occupying the same canopy layer so that rao’s index might actually prove significant if these layers are analyzed separately.

It would be great if you could add this element to the analysis. Even the SEM could be enriched accordingly.

I only report here some suggestions or examples of sentences that need to be revised concerning the first half of the manuscript but the same is true for the remaining part.

Finally,

The data made available only contains your plot level processed values used for correlations and SEM.

I t would be great if you could also add some other data for example the species list and the trait values used for each species. Even better it would be to have the species list per plot.

Line 28 still reads RATION and not RATIO

LINE 33 the TreeDivNet (Public https://treedivnet.ugent.be/publications.html ations – TreeDivNet) publishes extensively on Biodiversity and Ecosystem Function relationships please consider adding a refennce here for the list of pubblications you can find in the website. For example:

Enhancing Tree Performance Through Species Mixing: Review of a Quarter- Century of TreeDivNet Experiments Reveals Research Gaps and Practical Insights. Current Forestry Reports - https://doi.org/10.1007/s40725-023-00208-y

Or

LINE 34 PROMOTES not PROMOTED

LINE38 COMPLEMENTARITY not COMPLEMENTARY

LINE 44 you state “making them more likely to have a positive impact on ecosystem functionality” while it should be “making them more likely to include at least one species with high performance in the ecosystem function of interest”

LINES 59 to 62: The observed positive correlation between variations in individual tree size and biomass (or productivity) may be attributed to the niche complementarity effect, which posits that forest communities exhibiting greater structural complexity can enhance resource use efficiency, thereby increasing forest functioning [6,17,19,20].

Something wrong with this sentence: it speaks of the correlation between individual tree size and productivity, which is always true in any kind of context and thus it is not clear how it can be attributed to niche complementarity. Clearly something different has to be stated in the first prat of this sentence.

Line 104 The largest tall and diameter Highest and largest tree attained 25.6 m and 64.8 cm in diameter, respectively.

Line 117: replacing missing values can be tricky: please state for which traits and species the trait was not available and with what they were replaced. This is particularly important if one of the species for which you don’t have traits is one of the seven main species. Also not that instead of using the value of the closest species you could use the average of the genus.

Lines 142-144: something wrong here, it sounds as if you are looking for the correlation between the trait of one species and productivity which is not possible because there is no variance with one value, so it is not clear what has been done. Please clarify.

Line 173: “Within a 6-ha plot, stand productivity ranged from 0.53 to 2.82 Mg yr–1.” It is not clear her and in the sentences below if the data is actually referring to the 150 plots of the experiment. So you are saying that the least productive plot has 0.53 and the most productive has 2.82 Mg yr-1? Please clarify.

The values you give for each species are higher than the highest value reported above, so perhaps you are speaking of the productivity of the species across the 6ha experiment? It might help to report the productivity per hectare.

Lines 180: here you are reporting the correlation between the productivity of a plot and the MH of the tallest tree of the CWMmh? Please clarify.

Reviewer #2: There are still some typos/grammar errors that should be cleaned up, but the paper as a whole looks good.

7. PLOS authors have the option to publish the peer review history of their article (what does this mean? ). If published, this will include your full peer review and any attached files.

**Do you want your identity to be public for this peer review?** For information about this choice, including consent withdrawal, please see our Privacy Policy .

Reviewer #1: No

Reviewer #2: No

---

## [Author Response · Author response to Decision Letter 2]

23 Apr 2025

Line 28 still reads RATION and not RATIO

RE: Thank you for your suggestion.We have made changes in the article and see line 29 for the updated text.

Line 33 the TreeDivNet (Public https://treedivnet.ugent.be/publications.html ations – TreeDivNet) publishes extensively on Biodiversity and Ecosystem Function relationships please consider adding a refennce here for the list of pubblications you can find in the website. For example:

Enhancing Tree Performance Through Species Mixing: Review of a Quarter- Century of TreeDivNet Experiments Reveals Research Gaps and Practical Insights. Current Forestry Reports - https://doi.org/10.1007/s40725-023-00208-y

Or

RE: Thank you for your suggestion. We have re-inserted the relevant references through the channels you provided, see line 34 for the updated text

Line 34 PROMOTES not PROMOTED

RE: Thank you for your suggestion. I have revised it. See Line 35.

Line38 COMPLEMENTARITY not COMPLEMENTARY

RE: Thank you for your suggestion.We have corrected similar errors throughout the article

Line 44 you state “making them more likely to have a positive impact on ecosystem functionality” while it should be “making them more likely to include at least one species with high performance in the ecosystem function of interest”

RE: Thank you for your suggestion. After reviewing a large amount of literature, we found that the common expression is "the selection effect states that more diverse communities are likely to contain more species with high performance in ecosystem functions." See lines 41-42

Lines 59 to 62: The observed positive correlation between variations in individual tree size and biomass (or productivity) may be attributed to the niche complementarity effect, which posits that forest communities exhibiting greater structural complexity can enhance resource use efficiency, thereby increasing forest functioning [6,17,19,20].

Something wrong with this sentence: it speaks of the correlation between individual tree size and productivity, which is always true in any kind of context and thus it is not clear how it can be attributed to niche complementarity. Clearly something different has to be stated in the first prat of this sentence.

RE: Thank you for your suggestion. It was indeed an incorrect expression — it should not be "individual tree size," but rather "tree-size inequality" or "tree-size variability." We have corrected it accordingly. See lines 56-59

Line 104 The largest tall and diameter Highest and largest tree attained 25.6 m and 64.8 cm in diameter, respectively.

RE: Thank you for your suggestion. I have revised it. Changed “The maximum tree height and diameter at breast height (DBH, measured at 1.3 m height) reached 25.6 m and 64.8 cm, respectively”. See Lines 104-105

Line 117: replacing missing values can be tricky: please state for which traits and species the trait was not available and with what they were replaced. This is particularly important if one of the species for which you don’t have traits is one of the seven main species. Also not that instead of using the value of the closest species you could use the average of the genus.

RE: Thank you for your suggestion. In this study, the substituted plant functional trait values constituted only a minor proportion. We compared the results obtained by using similar genus values and those obtained by using the average genus value, and found that both approaches had negligible effects on the final outcome.

Lines 142-144: something wrong here, it sounds as if you are looking for the correlation between the trait of one species and productivity which is not possible because there is no variance with one value, so it is not clear what has been done. Please clarify.

RE: Thank you for your suggestion. I have revised it. Changed “Within the 6-ha plot, we analyzed the relative contribution of different species to overall productivity.”. See Lines 144-145

Line 173: “Within a 6-ha plot, stand productivity ranged from 0.53 to 2.82 Mg yr–1.” It is not clear her and in the sentences below if the data is actually referring to the 150 plots of the experiment. So you are saying that the least productive plot has 0.53 and the most productive has 2.82 Mg yr-1? Please clarify.

The values you give for each species are higher than the highest value reported above, so perhaps you are speaking of the productivity of the species across the 6ha experiment? It might help to report the productivity per hectare.

RE: Thank you for your suggestion. Here, we present the productivity values of each species and their respective proportions at the species level, which have been appropriately adjusted.See Lines 178-179. Regarding the stand productivity value you mentioned, it ranges from 0.53 to 2.82 Mg·yr-1. Specifically, on the 20×20 quadrat scale, the minimum quadrat productivity value is 0.53 Mg·yr-1, while the maximum quadrat productivity value is 2.82 Mg·yr-1.See Line 192

Lines 180: here you are reporting the correlation between the productivity of a plot and the MH of the tallest tree of the CWMmh? Please clarify.

RE: Thank you for your suggestion. I have revised it. Changed “The analysis of the relationship between functional trait values and productivity of each species in this plot showed that MH was significantly and positively correlated with productivity (R2=0.22, P�0.001; Figure 1d), whereas WD was negatively correlated with productivity (R2=0.05, P�0.05; Figure 1e). See Lines 184-185

---

## [Editor Report · Decision Letter 2]

28 Apr 2025

Functional trait identity regulates productivity better than tree diversity and structural complexity in subtropical mixed-species forests

PONE-D-24-26752R2

Dear Dr. Zhang,

We’re pleased to inform you that your manuscript has been judged scientifically suitable for publication and will be formally accepted for publication once it meets all outstanding technical requirements.

Kind regards,

Francesco Boscutti

Academic Editor

PLOS ONE

---

## [Editor Report · Acceptance letter]

PONE-D-24-26752R2

PLOS ONE

Dear Dr. Zhang,

I'm pleased to inform you that your manuscript has been deemed suitable for publication in PLOS ONE. Congratulations! Your manuscript is now being handed over to our production team.

Kind regards,

on behalf of

Dr. Francesco Boscutti

Academic Editor

PLOS ONE